# Female genital mutilation and safer sex negotiation among women in sexual unions in sub-Saharan Africa: Analysis of demographic and health survey data

Richard Gyan Aboagye[1,2], Bright Opoku Ahinkorah[3,4,5], Abdul-Aziz Seidu[5,6,7], James Boadu Frimpong[8,9], Collins Adu[10,11]*, John Elvis Hagan Jr.[8,12], Salma A. E. Ahmed[4], Sanni Yaya[13,14]

1 School of Population Health, University of New South Wales, Sydney, NSW, Australia, 2 Department of Family and Community Health, Fred N. Binka School of Public Health, University of Health and Allied Sciences, Ho, Ghana, 3 School of Clinical Medicine, University of New South Wales Sydney, Sydney, Australia, 4 School of Public Health, Faculty of Health, University of Technology Sydney, Sydney, Australia, 5 REMS Consultancy Services Limited, Sekondi-Takoradi, Western Region, Ghana, 6 College of Public Health, Medical and Veterinary Services, James Cook University, Townsville, Queensland, Australia, 7 Centre for Gender and Advocacy, Takoradi Technical University, Takoradi, Ghana, 8 Department of Health, Physical Education and Recreation, University of Cape Coast, Cape Coast, Ghana, 9 Department of Kinesiology, New Mexico State University, Las Cruces, NM, United States of America, 10 Department of Health Promotion, Education and Disability Studies, Kwame Nkrumah University of Science and Technology, Kumasi, Ghana, 11 Centre for Social Research in Health, University of New South Wales Sydney, Sydney, NSW, Australia, 12 Neurocognition and Action-Biomechanics-Research Group, Faculty of Psychology and Sport Sciences, Bielefeld University, Bielefeld, Germany, 13 School of International Development and Global Studies, University of Ottawa, Ottawa, Canada, 14 The George Institute for Global Health, Imperial College London, London, United Kingdom

* collinsadu80@yahoo.com

**Data Availability Statement:** https://dhsprogram.com/data/available-datasets.cfm.

## Abstract

### Background

The practice of female genital mutilation is associated with harmful social norms promoting violence against girls and women. Various studies have been conducted to examine the prevalence of female genital mutilation and its associated factors. However, there has been limited studies conducted to assess the association between female genital mutilation and markers of women's autonomy, such as their ability to negotiate for safer sex. In this study, we examined the association between female genital mutilation and women's ability to negotiate for safer sex in sub-Saharan Africa (SSA).

### Methods

We pooled data from the most recent Demographic and Health Surveys (DHS) conducted from 2010 to 2020. Data from a sample of 50,337 currently married and cohabiting women from eleven sub-Saharan African countries were included in the study. A multilevel binary logistic regression analysis was used to examine the association between female genital mutilation and women's ability to refuse sex and ask their partners to use condom. Adjusted

**Funding:** The author(s) received no specific funding for this work.

**Competing interests:** The authors have declared that no competing interests exist.

odds ratios (aORs) with a 95% confidence interval (CI) were used to present the findings of the logistic regression analysis. Statistical significance was set at p<0.05.

## Results

Female genital mutilation was performed on 56.1% of women included in our study. The highest and lowest prevalence of female genital mutilation were found among women from Guinea (96.3%) and Togo (6.9%), respectively. We found that women who had undergone female genital mutilation were less likely to refuse sex from their partners (aOR = 0.91, 95% CI = 0.86, 0.96) and ask their partners to use condoms (aOR = 0.82, 95% CI = 0.78, 0.86) compared to those who had not undergone female genital mutilation.

## Conclusion

Female genital mutilation hinders women's ability to negotiate for safer sex. It is necessary to implement health education and promotion interventions (e.g., decision making skills) that assist women who have experienced female genital mutilation to negotiate for safer sex. These interventions are crucial to enhance sexual health outcomes for these women. Further, strict enforcement of policies and laws aimed at eradicating the practice of female genital mutilation are encouraged to help contribute to the improvement of women's reproductive health.

## Background

Female genital mutilation (FGM) is a major public health concern and human rights issue affecting girls and women worldwide [1]. It is widely recognized as a grave violation of human rights, with detrimental effects on the physical and mental wellbeing of millions of girls and women. In addition, it places a significant burden on a country's financial resources [2]. FGM is highly prevalent in 30 countries, putting an estimated three million girls at risk of undergoing the practice annually [1, 2]. It is most prevalent in Africa but also occurs in Asia and other parts of Europe where there are communities with origins in FGM-practicing societies [3]. In some parts of Africa, FGM is deeply rooted in religious facets and social mores, with justification including the preservation of virginity, marriage requirements, cultural identity, hygiene, conjugal fidelity, honour, fertility beliefs, initiation rites, and notions of purity [2, 4, 5]. The World Health Organization (WHO) classifies FGM into four categories. Types 1 and 2, known as clitoridectomy and excision respectively, involves the partial or complete removal of the clitoris and labia. Type 3, known as infibulation, involves cutting and repositioning the labia to create a partial covering, sometimes requiring the stitching together of the tissues (this is the most extreme form of FGM). Type 4 involves the piercing or scraping the genitalia [2].

According to a UNICEF report [1], over 90% of FGM incidents are Types 1 (primarily clitoridectomy), 2 (excision), or 4 ("nicking" without flesh removed), with the remaining 10% (nearly 8 million women) being infibulated. The countries that practice infibulation the most are Djibouti, Eritrea, Ethiopia, Somalia, and Sudan. In West Africa (e.g Guinea, Mali, and Burkina Faso), the tendency is to remove flesh (clitoridectomy and/or excision) rather than suture the labia minora and/or majora together [1]. FGM has adverse consequences, including extreme pain, haemorrhage, infection, cyst and keloidal formation, sexual dysfunction, chronic pelvic infection, obstetric issues, and death in the worst cases [1, 3, 6].

In most countries in sub-Saharan Africa (SSA), the conventional social organization is generally patriarchal, with males dominating women [7]. Safer sex negotiation in sexual partnerships has several advantages, including a reduction in sexually transmitted infections (STIs) [8]. STIs, particularly HIV/AIDS, disproportionately affect women [9]. Women's control over their sexual lives plays a significant role in determining their vulnerability to STIs [10]. The Sustainable Development Goal (SDG) 5 [11, 12] focuses on gender equality and empowerment of girls and women, including improving women's ability to negotiate for safer sex. As a result, authorities, especially in low- and middle-income countries, are increasingly paying attention to issues related to women's sexual autonomy [7].

The study utilized the normative social influence theory which explains how people's behaviours are influenced by social norms [13]. Essentially, people conform to the social norms established within their community members and families to be accepted and to avoid marginalization or exclusion [14]. Research shows that people tend to act in accordance with established societal norms, and deviation from those norms can make it difficult or impossible to fit into the community [15]. A crucial aspect of the normative social influence theory is how norms are transmitted among males and females from childhood to adolescence and adulthood [13]. These norms become deeply ingrained by adulthood, making it challenging to break away from them. Therefore, women who have undergone FGM acquire a certain identity within the community that makes them less assertive in negotiating for safer sex compared to their peers who have not undergone FGM [16].

Previous studies have identified factors such as place of residence, marital status, age, and educational level to be associated with safer sex negotiation [17–20]. Although FGM is linked to harmful social norms that contribute to violence against girls and women, to the best of our knowledge, there is no research on the association between FGM and women's ability to negotiate safer sex in SSA. In this study, we examined the association between FGM and safer sex negotiation among women in sexual unions in SSA.

## Methods

### Data source and study design

We analyzed cross-sectional data from the most recent Demographic and Health Surveys (DHS) conducted in 11 countries in SSA (Table 1). We included countries with the most recent datasets from 2010 to 2020. Only countries with variables on FGM, safer sex

**Table 1. Description of the study sample.**

| S/N Country | Survey year | Weighted N | Weighted % |
|---|---|---|---|
| 1. Burkina Faso | 2010 | 9,898 | 19.7 |
| 2. Ethiopia | 2016 | 2,541 | 5.0 |
| 3. Gambia | 2019–20 | 2,139 | 4.2 |
| 4. Guinea | 2018 | 2,577 | 5.1 |
| 5. Kenya | 2014 | 6,702 | 13.3 |
| 6. Liberia | 2019–20 | 2,142 | 4.3 |
| 7. Mali | 2018 | 2,279 | 4.5 |
| 8. Nigeria | 2018 | 8,443 | 16.8 |
| 9. Sierra Leone | 2019 | 5,745 | 11.4 |
| 10. Senegal | 2010–11 | 5,122 | 10.2 |
| 11. Togo | 2013–14 | 2,749 | 5.5 |
| **All countries** | **2010–2020** | **50,337** | **100.0** |

negotiation, and other variables considered in this study were included. In the DHS, respondents were selected using a two-stage cluster sampling method [21]. Structured questionnaires were used to collect data from the respondents on health and social indicators such as FGM, safer sex negotiation, and reproductive health and rights [21]. For this study, we included 50,337 currently married and cohabiting women of reproductive age (15–49 years). The datasets are freely accessible via this link: https://dhsprogram.com/data/available-datasets.cfm.

## Variables

**Outcome variable.** We considered two outcome variables: women's ability to refuse sex and ask their partners to use condom. The respondents were questioned if they could refuse sex with their partners. Additionally, they were asked if they could ask their partners to use condoms. The response options for each of the two variables were the same: "no", "yes", and "don't know/not sure/depends". We used the definite response options in the final analysis. Hence, women whose response options were "don't know/not sure/depends" were excluded. We maintained the dichotomized responses (0 = no and 1 = yes) in the final analysis. The coding and categorization were based on previous studies [20, 22–25].

**Key explanatory variable.** The main explanatory variable in this study was FGM, which was derived from the question "Have you yourself ever been circumcised?" The response options were "yes" and "no". We recoded these dichotomized responses, assigning 0 for "no" and 1 for "yes". This coding approach was chosen based on existing literature that used the DHS dataset [26].

**Covariates.** The study included 14 variables as covariates. These variables grouped as individual and contextual level variables, were selected based on their significant association with safer sex negotiation in previous studies [20, 22, 24, 27, 28].

*Individual level variables.* From the DHS, we used the existing coding for women's age (15–19; 20–24; 25–29; 30–34; 35–39; 40–44; and 45–49), educational level of the women and their partners (no education; primary; secondary; and higher), and current working status (no/yes). The marital status of respondents was recoded as "married" and "cohabiting". Partners age was coded as 15–24; 25–34; 35–44; and 45+. Religion was recoded as "Christianity"; "Islam"; "African Traditional"; "No religion"; and "Others". Frequency of listening to radio, frequency of watching television, and frequency of reading newspaper or magazines were all recoded as "not at all"; "less than once a week"; and "at least once a week". Comprehensive HIV/AIDS knowledge was categorized into "no" and "yes".

*Contextual variables.* Wealth index (poorest, poorer, middle, richer, and richest), place of residence (urban and rural), and studied countries were the contextual variables considered in our study.

## Statistical analyses

All analyses were performed using Stata software version 16.0 (Stata Corporation, College Station, TX, USA). First, we used percentages to present the prevalence of FGM, women's ability to refuse sex, and women's ability to ask their partners to use a condom. We used cross-tabulations to examine the distribution of the outcome variables across FGM and the covariates. The Pearson's chi-square test of independence was used to determine the variables with a significant association with the outcome variables. After this procedure, collinearity was checked among the studied variables using the variance inflation factor. The results showed that the minimum, maximum, and mean variance inflation factors were 1.00, 3.74, and 1.84, respectively. Hence, there was no evidence of high collinearity among the variables included in the study. A multilevel binary logistic regression analysis was used to examine the association

between FGM and women's ability to refuse sex and ask their partners to use condom. Four models were built to examine this association (Model O, I, II, and III). Model O was an empty model with no explanatory variable. We included FGM and individual-level variables in Model I. Model II was fitted to contain the contextual-level variables. Model III had all the explanatory variables (FGM and covariates). However, the results for Models 0 and III are presented in Table 3, with the complete tables containing the four models attached as a supplementary file. A Stata command "melogit" was used to fit the models. Akaike's Information Criterion (AIC) test was used to compare the fitness of the model with the last model being the best-fitted model. The first model was selected because it is the empty model and subsequently compared its results with the last model which had the highest log-likelihood, and the least AIC values. Also, the association between FGM and the outcome variables per country was examined (Table 4). Adjusted odds ratios (aOR) with 95% confidence interval (CI) were used to present the results of the regression analyses. Statistical significance was set at $p < 0.05$ in the chi-square test and regression analyses. All the analyses were weighted. We weighted the data at the country level by appending the dataset for all the countries. In doing this, we applied the women's sample weights and strata. First, the women's weighting variable (v005) was divided by 1000000 to generate a new variable called "= v005_pw". Subsequently, we de-normalized the data using the command: v005 × (total female population 15–49 in the country)/ (total number of women 15–49 interviewed in the survey), and then re-normalized so that in the pooled sample the average is 1. We then appended the cleaned country-level dataset for the final analysis.

### Ethical consideration

We did not seek ethical clearance for this study since the DHS dataset is available in the public domain. Detailed information about the DHS data usage and ethical standards are available at http://goo.gl/ny8T6X.

## Results

### Prevalence of female genital mutilation and safer sex negotiation among women in sub-Saharan Africa

Fig 1 depicts the prevalence of FGM and safer sex negotiation per country. FGM was found to be prevalent in 56.1% of the women included in the study. Guinea had the highest prevalence (96.3%), while Togo had the lowest (6.9%). The prevalence of women's ability to refuse sex and ask for condom use were 60.0% and 50.9%, respectively. Liberia had the highest percentage of women who said they were able to refuse sex (85.4%), while Mali had the lowest (30%). Kenya had the highest percentage of women who could ask their partners to use condoms (79.4%), while Guinea had the lowest (35.7%).

### Distribution of safer sex negotiation across female genital mutilation and covariates

Table 2 shows the results of the distribution of safer sex negotiation (refuse sex and ask for condom use) across FGM as well as the association between the explanatory variables and safer sex negotiation. Among women who had undergone FGM, the prevalence of women's ability to refuse sex and ask their partner to use condom were 56.8% and 44.6%, respectively. FGM status was associated with women's ability to refuse sex ($p < 0.001$) and to ask partners to use condoms ($p < 0.001$). Additionally, all the covariates had a statistically significant association with the outcome variables ($p < 0.001$).

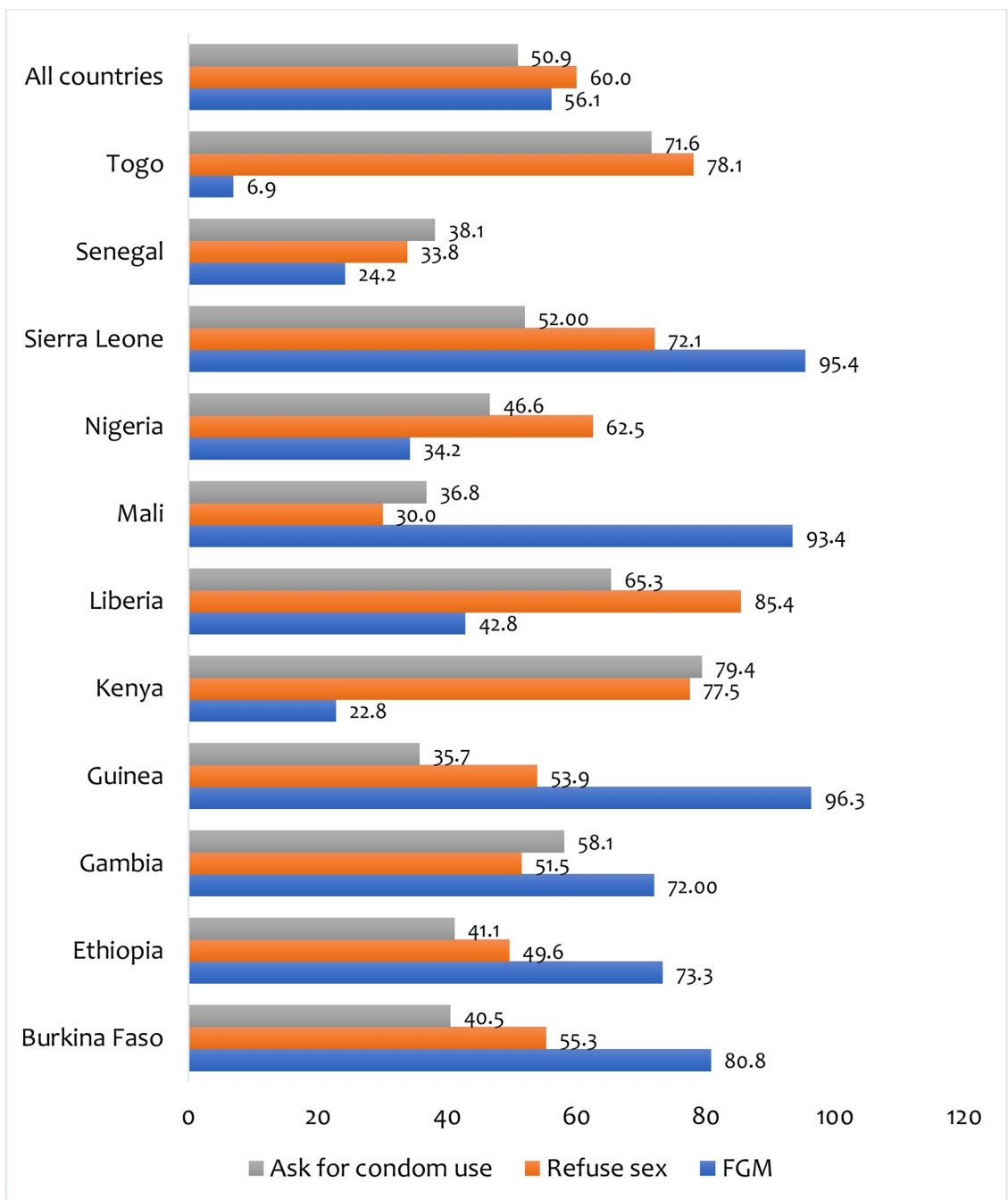

**Fig 1. Prevalence of female genital mutilation and safer sex negotiation among women in sub-Saharan Africa.**

## Association between female genital mutilation and safer sex negotiation

Table 3 presents the results of the association between FGM and safer sex negotiation among women in SSA. Women with a history of FGM had lower odds of refusing sexual intercourse from their partners compared to those who had not experienced FGM (aOR = 0.91, 95% CI = 0.86, 0.96). Women who had experienced FGM were also less likely to request their partners to use condoms during sex compared to those with no history of FGM (aOR = 0.82, 95%

**Table 2. Distribution of safer sex negotiation across female genital mutilation and covariates.**

| Variables | Weighted N | Weighted % | Refuse sex Yes (%) | p-value | Ask for condom use Yes (%) | p-value |
|---|---|---|---|---|---|---|
| **Female genital mutilation** | | | | <0.001 | | <0.001 |
| Not undergone FGM | 22,082 | 43.9 | 64.2 | | 59.0 | |
| Undergone FGM | 28,255 | 56.1 | 56.8 | | 44.6 | |
| **Women's age (years)** | | | | <0.001 | | <0.001 |
| 15–19 | 2,915 | 5.8 | 54.0 | | 45.7 | |
| 20–24 | 8,392 | 16.7 | 59.7 | | 52.3 | |
| 25–29 | 11,171 | 22.2 | 61.3 | | 54.9 | |
| 30–34 | 9,482 | 18.8 | 60.4 | | 52.2 | |
| 35–39 | 8,262 | 16.4 | 61.1 | | 50.3 | |
| 40–44 | 5,750 | 11.4 | 60.3 | | 47.6 | |
| 45–49 | 4,365 | 8.7 | 58.4 | | 43.9 | |
| **Marital status** | | | | <0.001 | | <0.001 |
| Married | 46,728 | 92.8 | 58.4 | | 49.2 | |
| Cohabiting | 3,609 | 7.2 | 80.6 | | 72.6 | |
| **Women's educational level** | | | | <0.001 | | <0.001 |
| No education | 25,534 | 50.7 | 49.4 | | 35.2 | |
| Primary | 11,111 | 22.1 | 65.3 | | 61.2 | |
| Secondary | 10,636 | 21.1 | 73.4 | | 69.6 | |
| Higher | 3,056 | 6.1 | 83.4 | | 79.5 | |
| **Partner's educational level** | | | | <0.001 | | <0.001 |
| No education | 23,275 | 46.3 | 48.5 | | 34.9 | |
| Primary | 9,325 | 18.5 | 64.0 | | 58.4 | |
| Secondary | 12,343 | 24.5 | 71.5 | | 66.3 | |
| Higher | 5,395 | 10.7 | 76.8 | | 71.9 | |
| **Partner's age (years)** | | | | <0.001 | | <0.001 |
| 15–24 | 1,723 | 3.4 | 63.8 | | 56.1 | |
| 25–34 | 13,795 | 27.4 | 63.4 | | 57.6 | |
| 35–44 | 16,543 | 32.9 | 61.3 | | 53.4 | |
| 45+ | 18,276 | 36.3 | 56.0 | | 43.1 | |
| **Current working status** | | | | <0.001 | | <0.001 |
| Not working | 15,675 | 31.1 | 54.2 | | 46.8 | |
| Working | 34,662 | 68.9 | 62.7 | | 52.7 | |
| **Religion** | | | | <0.001 | | <0.001 |
| Christianity | 19,129 | 38.0 | 75.8 | | 67.9 | |
| Islamic | 29,571 | 58.8 | 49.7 | | 40.5 | |
| African Traditional | 1,161 | 2.3 | 61.4 | | 32.7 | |
| No religion | 421 | 0.8 | 63.0 | | 51.9 | |
| Others | 55 | 0.1 | 79.4 | | 80.3 | |
| **Comprehensive HIV and AIDS knowledge** | | | | <0.001 | | <0.001 |
| No | 27,789 | 55.2 | 54.9 | | 44.2 | |
| Yes | 22,548 | 44.8 | 66.3 | | 59.1 | |
| **Frequency of reading newspaper/magazine** | | | | <0.001 | | <0.001 |
| Not at all | 42,913 | 85.3 | 57.5 | | 46.7 | |
| Less than once a week | 4,391 | 8.7 | 74.6 | | 73.3 | |
| At least once a week | 3,033 | 6.0 | 75.0 | | 77.5 | |
| **Frequency of listening to radio** | | | | <0.001 | | <0.001 |
| Not at all | 15,949 | 31.7 | 56.5 | | 42.6 | |

(*Continued*)

**Table 2.** (Continued)

| Variables | Weighted N | Weighted % | Refuse sex Yes (%) | p-value | Ask for condom use Yes (%) | p-value |
|---|---|---|---|---|---|---|
| Less than once a week | 10,928 | 21.7 | 60.1 | | 50.6 | |
| At least once a week | 23,459 | 46.6 | 62.4 | | 56.7 | |
| **Frequency of watching television** | | | | <0.001 | | <0.001 |
| Not at all | 26,827 | 53.3 | 57.4 | | 42.7 | |
| Less than once a week | 7,477 | 14.8 | 60.6 | | 54.5 | |
| At least once a week | 16,033 | 31.9 | 64.2 | | 62.9 | |
| **Wealth index** | | | | <0.001 | | <0.001 |
| Poorest | 8,718 | 17.3 | 53.0 | | 36.5 | |
| Poorer | 9,468 | 18.8 | 55.4 | | 41.7 | |
| Middle | 9,794 | 19.5 | 56.2 | | 47.0 | |
| Richer | 10,618 | 21.1 | 62.2 | | 56.1 | |
| Richest | 11,738 | 23.3 | 70.2 | | 67.6 | |
| **Residence** | | | | <0.001 | | <0.001 |
| Urban | 19,636 | 39.0 | 67.9 | | 63.3 | |
| Rural | 30,701 | 61.0 | 55.0 | | 43.0 | |

CI = 0.78, 0.86). With the covariates, cohabiting women were more likely to negotiate for safer sex compared to married women. Higher educational status of women and their partners enhanced the likelihood of safer sex negotiation. Compared to women who were not exposed to television, radio, or newspapers/magazines, those who were exposed had higher odds of negotiating for safer sex. In comparison to non-working women, employed women were more inclined to negotiate for safer sex. Compared to women without comprehensive HIV and AIDS knowledge, those with comprehensive HIV and AIDS knowledge were more likely to negotiate for safer sex. Women in rural areas had lower odds in negotiating for safer sex compared to women in urban areas. The complete results for the association between FGM and women's ability to negotiate for condom use as well as women's ability to refuse partner's sex in SSA are provided as S1 and S2 Tables in S1 File.

### Association between female genital mutilation and safer sex negotiation by country

Table 4 present the results of the association between FGM and safer sex negotiation segregated by country. We found that women from Burkina Faso, Guinea, and Kenya who had undergone FGM were less likely to refuse sexual intercourse with their partners. Also, the odds of women's ability to ask their partner to use a condom was lower among women with a history of FGM from Burkina Faso, Guinea, Kenya, and Togo.

### Discussion

The study examined the association between FGM and safer sex negotiation among women in sexual unions in SSA. The findings indicate that women who have undergone FGM are less likely to refuse sexual intercourse and request condom use from their partners. These findings are consistent with other recent studies which have also identified FGM as a significant factor that influences women's sexual behaviours in various African countries [13, 29]. The practice of FGM, primarily used in certain societies to regulate sexual desire and ensure the virginity and chastity of females until marriage, can have an impact on women's capacity to negotiate safe sex [13]. Chai et al. [13] suggests that women within FGM practicing communities may

**Table 3. Mixed effects analysis female genital mutilation and safer sex negotiation among women in sub-Saharan Africa.**

| Variables | Ability to refuse partner's sex | | Ability to ask partner to use condom | |
|---|---|---|---|---|
| | Model O | Model III aOR [95% CI] | Model O | Model III aOR [95% CI] |
| **Fixed effect results** | | | | |
| **Female genital mutilation** | | | | |
| Not undergone FGM | | Reference category | | Reference category |
| Undergone FGM | | 0.91*** [0.86, 0.96] | | 0.82*** [0.78, 0.86] |
| **Women's age (years)** | | | | |
| 15–19 | | Reference category | | Reference category |
| 20–24 | | 1.04 [0.95,1.15] | | 1.05 [0.96,1.16] |
| 25–29 | | 1.04 [0.94,1.15] | | 1.07 [0.97,1.18] |
| 30–34 | | 1.05 [0.94,1.17] | | 1.07 [0.96,1.19] |
| 35–39 | | 1.09 [0.98,1.22] | | 1.04 [0.93,1.16] |
| 40–44 | | 1.12 [0.99,1.26] | | 1.04 [0.92,1.17] |
| 45–49 | | 1.05 [0.92,1.19] | | 0.94 [0.83,1.06] |
| **Marital status** | | | | |
| Married | | Reference category | | Reference category |
| Cohabiting | | 1.33*** [1.20,1.46] | | 1.40*** [1.28,1.53] |
| **Women's educational level** | | | | |
| No education | | Reference category | | Reference category |
| Primary | | 1.24*** [1.17,1.31] | | 1.37*** [1.29,1.45] |
| Secondary | | 1.47*** [1.37,1.59] | | 1.59*** [1.49,1.71] |
| Higher | | 2.03*** [1.78,2.33] | | 2.03*** [1.78,2.31] |
| **Current working status** | | | | |
| Not working | | Reference category | | Reference category |
| Working | | 1.06* [1.01,1.11] | | 1.17*** [1.12,1.23] |
| **Religion** | | | | |
| Christianity | | Reference category | | Reference category |
| Islamic | | 0.55*** [0.52,0.59] | | 0.62*** [0.59,0.66] |
| African Traditional | | 0.93 [0.82,1.07] | | 0.52*** [0.45,0.60] |
| No religion | | 0.64*** [0.52,0.79] | | 0.62*** [0.50,0.77] |
| Others | | 1.52 [0.73,3.16] | | 2.28* [1.07,4.87] |
| **Comprehensive HIV and AIDS knowledge** | | | | |
| No | | Reference category | | Reference category |
| Yes | | 1.18*** [1.14,1.24] | | 1.30*** [1.24,1.35] |
| **Partner's age (years)** | | | | |
| 15–24 | | Reference category | | Reference category |
| 25–34 | | 0.96 [0.85,1.09] | | 0.92 [0.82,1.04] |
| 35–44 | | 0.91 [0.80,1.04] | | 0.84** [0.74,0.95] |
| 45+ | | 0.85* [0.74,0.97] | | 0.69*** [0.60,0.79] |
| **Partner's educational level** | | | | |
| No education | | Reference category | | Reference category |
| Primary | | 1.14*** [1.07,1.22] | | 1.24*** [1.16,1.32] |
| Secondary | | 1.19*** [1.12,1.27] | | 1.40*** [1.32,1.49] |
| Higher | | 1.21*** [1.10,1.33] | | 1.43*** [1.31,1.57] |
| **Frequency of watching television** | | | | |
| Not at all | | Reference category | | Reference category |
| Less than once a week | | 1.06 [1.00,1.13] | | 1.20*** [1.12,1.27] |
| At least once a week | | 1.15*** [1.08,1.23] | | 1.25*** [1.18,1.34] |

*(Continued)*

**Table 3.** (Continued)

| Variables | Ability to refuse partner's sex | | Ability to ask partner to use condom | |
|---|---|---|---|---|
| | Model O | Model III aOR [95% CI] | Model O | Model III aOR [95% CI] |
| **Frequency of listening to radio** | | | | |
| Not at all | | Reference category | | Reference category |
| Less than once a week | | 1.15*** [1.09,1.23] | | 1.06 [1.00,1.12] |
| At least once a week | | 1.20*** [1.14,1.26] | | 1.15*** [1.09,1.21] |
| **Frequency of reading newspaper/magazine** | | | | |
| Not at all | | Reference category | | Reference category |
| Less than once a week | | 1.16*** [1.06,1.27] | | 1.29*** [1.18,1.40] |
| At least once a week | | 1.10 [0.98,1.23] | | 1.32*** [1.18,1.47] |
| **Wealth index** | | | | |
| Poorest | | Reference category | | Reference category |
| Poorer | | 1.01 [0.95,1.07] | | 1.07* [1.00,1.14] |
| Middle | | 0.99 [0.93,1.06] | | 1.13*** [1.06,1.21] |
| Richer | | 1.03 [0.96,1.11] | | 1.25*** [1.16,1.35] |
| Richest | | 1.06 [0.96,1.16] | | 1.38*** [1.26,1.51] |
| **Residence** | | | | |
| Urban | | Reference category | | Reference category |
| Rural | | 0.89*** [0.84,0.94] | | 0.88*** [0.83,0.93] |
| **Countries** | | | | |
| Burkina Faso | | Reference category | | Reference category |
| Ethiopia | | 0.73*** [0.65,0.81] | | 0.73*** [0.65,0.81] |
| Gambia | | 0.89* [0.80,1.00] | | 1.85*** [1.65,2.07] |
| Guinea | | 0.93 [0.84,1.02] | | 0.71*** [0.64,0.79] |
| Kenya | | 1.13* [1.03,1.25] | | 1.76*** [1.60,1.94] |
| Liberia | | 2.53*** [2.20,2.90] | | 0.99 [0.88,1.11] |
| Mali | | 0.34*** [0.30,0.38] | | 0.75*** [0.68,0.84] |
| Nigeria | | 1.04 [0.96,1.13] | | 0.74*** [0.68,0.80] |
| Sierra Leone | | 2.21*** [2.04,2.40] | | 1.51*** [1.40,1.64] |
| Senegal | | 0.35*** [0.32,0.38] | | 0.70*** [0.64,0.77] |
| Togo | | 1.44*** [1.28,1.61] | | 1.67*** [1.49,1.87] |
| **Random effect results** | | | | |
| Primary Sampling Unit variance (95% CI) | 0.41 [0.35, 0.47] | 0.20 [0.17, 0.23] | 0.60 [0.52, 0.68] | 0.22 [0.18, 0.26] |
| Intra-Class Correlation Coefficient | 0.11 | 0.06 | 0.15 | 0.06 |
| Likelihood ratio Test | 1242.69 (<0.001) | 694.55 (<0.001) | 1862.62 (<0.001) | 735.93 (<0.001) |
| Wald chi-square | Reference | 5811.30*** | Reference | 6552.77*** |
| **Model fitness** | | | | |
| Log-likelihood | -33288.29 | -29813.45 | -33959.44 | -30050.18 |
| Akaike's Information Criterion | 66580.59 | 59718.91 | 67922.87 | 60192.35 |
| Sample size | 50,337 | 50,337 | 50,337 | 50,337 |
| Number of clusters | 1,608 | 1,608 | 1,608 | 1,608 |

aOR = adjusted odds ratios; CI = Confidence Interval

* $p < 0.05$

** $p < 0.01$

*** $p < 0.001$

**Table 4. Association between female genital mutilation and safer sex negotiation by country.**

| Country | Ability to refuse partner's sex aOR [95% CI] | Ability to ask partner to use condom aOR [95% CI] |
|---|---|---|
| Burkina Faso | 0.86* [0.75, 0.99] | 0.81** [0.70, 0.93] |
| Ethiopia | 0.81 [0.64, 1.02] | 0.85 [0.66, 1.09] |
| Gambia | 1.17 [0.91, 1.50] | 0.85 [0.66, 1.10] |
| Guinea | 0.46** [0.27, 0.78] | 0.25*** [0.14, 0.44] |
| Kenya | 0.80** [0.68, 0.93] | 0.64*** [0.55, 0.75] |
| Liberia | 1.03 [0.78, 1.37] | 0.97 [0.78, 1.19] |
| Mali | 0.79 [0.52, 1.19] | 1.53 [0.98, 2.40] |
| Nigeria | 0.89 [0.77, 1.02] | 0.88 [0.77, 1.01] |
| Sierra Leone | 0.71 [0.47, 1.05] | 1.23 [0.87, 1.74] |
| Senegal | 1.19 [0.99, 1.43] | 0.94 [0.79, 1.14] |
| Togo | 1.01 [0.70, 1.44] | 0.62** [0.44, 0.87] |

Adjusted for the covariates; aOR = adjusted odds ratios; CI = Confidence Interval

* $p < 0.05$

** $p < 0.01$

*** $p < 0.001$

face stigma when expressing their sexual needs due to taboos around sexuality, sexual morals, and sociocultural expectations. This assertion aligns with the normative social influence theory and previous research indicating that in many of these communities, FGM is viewed as a cornerstone of moral virtue, making women who initiate sexual acts or negotiate during sex appear promiscuous [30]. Consequently, these social dynamics can limit women's sexual autonomy. However, factors such as high educational status, lack of financial dependence, access to resources and information can enhance women's autonomy, thereby increasing their ability to negotiate safer sexual behaviours.

Country-specific variations were observed, with the lower odds of refusing partner sex among women who have undergone FGM in Burkina Faso, Guinea, and Kenya. Similarly, women who have undergone FGM in Togo were less likely to ask their partners to use a condom. These findings are consistent with the findings of earlier studies conducted in Kenya [13] and the United States [29], further supporting the influence of FGM on women's sexual behaviour and autonomy. Given these findings, efforts should be directed towards the elimination of violence against women and girls in SSA, as well as promoting of family planning using a holistic approach.

Educated women are more inclined to weigh the advantages over the disadvantages when making decisions regarding their health [31]. Prior studies have suggested that better education for women and girls may reduce FGM and that educated women have economic power and can choose whether to undergo FGM [31, 32]. We also found that women with higher educational levels were more likely to deny sexual intercourse from their partners and request that they use a condom. These findings are consistent with those of prior studies [12, 25]. Women with higher levels of education are better informed when it comes to making important decisions about their sexual lives and health, which increases their sexual autonomy [33]. Women with higher education may have greater financial empowerment, making them more likely to negotiate for safer sex with their partners [12].

However, women who were cohabiting were also more likely to decline sex from their partners and encourage condom use, consistent with the findings of previous study [34]. One possible explanation for this finding is that cohabiting women may be more autonomous in their

sexual decisions, making them more likely to deny sex from their partners and ask them to use condoms. It is also possible that cohabiting women were shielding themselves from the stigma and guilt associated with being pregnant outside of marriage, making them more inclined to negotiate for safer sex from their intimate partners [34]. Nevertheless, women with partners who have higher levels of education were more likely to refuse intercourse and request condom use. This may be because women with more educated partners understand the importance of respecting their female partners' sexual life decisions about their sexual lives, as well as the potential consequences of going against their partner's wishes. This understanding increases their chances of negotiating for safer sex.

Women who had comprehensive HIV and AIDS knowledge were more likely to deny sex from their partners and request that they use a condom. The current study's findings are consistent with those of earlier studies [23, 34, 35]. Women who have a thorough understanding of HIV and AIDS could be aware of the repercussions of their behaviours in connection to their sexual lives, which could improve their sexual autonomy [23, 35].

Women who were exposed to mass media (television, radio, newspapers, or magazines) were more likely to refuse sex from their partners and request condom use. This finding is consistent with previous studies [22, 36–38]. The positive association between exposure to mass media and sexual autonomy [36, 37, 39] could be due to the information received about the adverse effects of not practicing safe sex, empowering them to negotiate for safer sex.

Other results showed that women living in rural areas were less likely to refuse sexual intercourse and request condom use from their partners. This aligns with a study conducted in SSA that found that women in rural areas had less autonomy in making informed decisions, such as negotiating for safer sex [12]. Lack of comprehensive sexual education may contribute to this lack of sexual autonomy among rural women [12].

## Strengths and limitations

The study has several strengths. The use of DHS data and robust statistical procedures support the reliability of our findings. Additionally, our findings bridge gaps in the current research on the association between FGM and safer sex negotiation. However, it is important to acknowledge the study's limitations. FGM was self-reported, likely leading to under-reporting. Furthermore, the cross-sectional design makes it difficult to establish causation between FGM and negotiation for safer sex. Also, the statistical analysis did not consider the role of social norms, despite the discussion of their connection to FGM and safer sex negotiation. Therefore, caution should be exercised when interpreting the study's findings. It is also important to note that our inferences are based on data from the standard DHS.

## Conclusion

The study findings highlight that women who have undergone FGM are less likely to negotiate for safer sex. This observation emphasizes the need for planned health education and promotion interventions that support these women in negotiating safer sex. Policymakers should prioritize the development and implementation of specific interventions aimed at preventing FGM, which is associated with adverse sexual and reproductive health outcomes.

## Supporting information

**S1 File. Supporting information containing S1 and S2 Tables.**
(DOCX)

## Author Contributions

**Conceptualization:** Richard Gyan Aboagye, Bright Opoku Ahinkorah, Abdul-Aziz Seidu.

**Formal analysis:** Richard Gyan Aboagye, Bright Opoku Ahinkorah, Abdul-Aziz Seidu.

**Methodology:** Bright Opoku Ahinkorah, Abdul-Aziz Seidu.

**Supervision:** Bright Opoku Ahinkorah, Abdul-Aziz Seidu.

**Writing – original draft:** Richard Gyan Aboagye, Bright Opoku Ahinkorah, Abdul-Aziz Seidu, James Boadu Frimpong, Collins Adu, John Elvis Hagan Jr., Salma A. E. Ahmed, Sanni Yaya.

**Writing – review & editing:** Richard Gyan Aboagye, Bright Opoku Ahinkorah, Abdul-Aziz Seidu, James Boadu Frimpong, Collins Adu, John Elvis Hagan Jr., Salma A. E. Ahmed, Sanni Yaya.

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
