## [Decision Letter · Decision Letter 0]

31 May 2022

PONE-D-21-27783Female genital mutilation and safer sex negotiation among women in sexual unions in sub-Saharan Africa: Analysis of Demographic and Health Survey dataPLOS ONE

Dear Dr. ADU,

Thank you for submitting your manuscript to PLOS ONE. After careful consideration, we feel that it has merit but does not fully meet PLOS ONE’s publication criteria as it currently stands. Therefore, we invite you to submit a revised version of the manuscript that addresses the points raised during the review process. Please note that we have only been able to secure a single reviewer to assess your manuscript. We are issuing a decision on your manuscript at this point to prevent further delays in the evaluation of your manuscript. Please be aware that the editor who handles your revised manuscript might find it necessary to invite additional reviewers to assess this work once the revised manuscript is submitted. However, we will aim to proceed on the basis of this single review if possible. 

Your manuscript has been assessed by an expert reviewer, whose comments are appended below. The reviewer has highlighted concerns about several aspects of the statistical analysis and discussion. Please ensure you respond to each point carefully in your response to reviewers document, and modify your manuscript accordingly.

We look forward to receiving your revised manuscript.

Kind regards,

Joseph Donlan

Editorial Office

PLOS ONE

Journal Requirements:

2. We note you have included a table to which you do not refer in the text of your manuscript. Please ensure that you refer to Table 3 in your text; if accepted, production will need this reference to link the reader to the Table.

Reviewers' comments:

Reviewer's Responses to Questions

**Comments to the Author**

1. Is the manuscript technically sound, and do the data support the conclusions?

Reviewer #1: Partly

2. Has the statistical analysis been performed appropriately and rigorously? 

Reviewer #1: No

3. Have the authors made all data underlying the findings in their manuscript fully available?

Reviewer #1: Yes

4. Is the manuscript presented in an intelligible fashion and written in standard English?

Reviewer #1: Yes

5. Review Comments to the Author

Reviewer #1: The authors have tackled a very important societal practice that has short and long term physical and psychological impact on women and girls. Their use of multi-country DHS dataset is also commendable as it presents an opportunity to understand how widespread the FGM practice is and also examine its impact in a diverse setting. However, the authors have not done a robust analysis of the dataset. Below are some concerns and suggestions.

The main concern is that the authors have not discussed the underlying mechanisms and linkages between FGM and other socio-demographic factors, considering that FGM is more likely to be done before marriage (<15yrs). The paper would benefit from some in-depth discussion of the link between FGM and the outcomes as well as link between FGM and socio-demographic factors. Two-thirds of the discussion is on the association between the outcomes and socio-demographic factors, with little focus on FGM (as their main exposure), which is the focus of the paper.

The other concern is on the analysis procedure. Much as it is important to look at the overall association between FGM and the outcomes, the analysis would have benefitted from individual country estimates, considering the wide variation in FGM (6.9% in Togo vs 96.3 % in Guinea) as well as safe sex practices across settings (30% in Mali vs 85% in Liberia). It is not clear if the authors examined the heterogeneity in the both the outcome and main exposure across the countries and how this was accounted for in the pooled analysis. It would important to examine how the risk of unsafe sex varies across the different settings.

In the data analysis section, the authors outlined four models they fitted but do not explain to the reader the criteria used in selecting the two models they present in table 3.

6. PLOS authors have the option to publish the peer review history of their article (what does this mean?). If published, this will include your full peer review and any attached files.

Reviewer #1: No

---

## [Author Response · Author response to Decision Letter 0]

10 Jul 2022

COMMENTS

Reviewer #1: The authors have tackled a very important societal practice that has short and long term physical and psychological impact on women and girls. Their use of multi-country DHS dataset is also commendable as it presents an opportunity to understand how widespread the FGM practice is and also examine its impact in a diverse setting. However, the authors have not done a robust analysis of the dataset. Below are some concerns and suggestions.

Our response: Thank you. We have addressed all the comments raised.

The main concern is that the authors have not discussed the underlying mechanisms and linkages between FGM and other socio-demographic factors, considering that FGM is more likely to be done before marriage (<15yrs). The paper would benefit from some in-depth discussion of the link between FGM and the outcomes as well as link between FGM and socio-demographic factors. Two-thirds of the discussion is on the association between the outcomes and socio-demographic factors, with little focus on FGM (as their main exposure), which is the focus of the paper.

Our response: Thank you very much. The discussion section has been revised to provide more information on FGM and safer sex negotiation, FGM and socio-demographic factors, and safer sex negotiation and socio-demographic factors.

The other concern is on the analysis procedure. Much as it is important to look at the overall association between FGM and the outcomes, the analysis would have benefitted from individual country estimates, considering the wide variation in FGM (6.9% in Togo vs 96.3 % in Guinea) as well as safe sex practices across settings (30% in Mali vs 85% in Liberia). It is not clear if the authors examined the heterogeneity in the both the outcome and main exposure across the countries and how this was accounted for in the pooled analysis. It would important to examine how the risk of unsafe sex varies across the different settings.

Our response: Thank you. We have provided a new result, which examined the association between FGM and safer sex negotiation by country (Table 4).

In the data analysis section, the authors outlined four models they fitted but do not explain to the reader the criteria used in selecting the two models they present in table 3.

Our response: Thank you. We have attached the complete tables containing the four models as supplementary file. We presented the results for only the two models because, the last models were the best-fitted models based on the lowest value of the Akaike’s Information Criterion and the highest value of the log-likelihood. Please refer to the supplementary file for detailed information.

---

## [Decision Letter · Decision Letter 1]

13 Sep 2023

PONE-D-21-27783R1Female genital mutilation and safer sex negotiation among women in sexual unions in sub-Saharan Africa: Analysis of Demographic and Health Survey dataPLOS ONE

Dear Dr. Adu,

Thank you for submitting your manuscript to PLOS ONE. After careful consideration, we feel that it has merit but does not fully meet PLOS ONE’s publication criteria as it currently stands. Therefore, we invite you to submit a revised version of the manuscript that addresses the points raised during the review process.

**After reading the revised version of the manuscript and based on the comments provided by Reviewer 2, I believe that the paper is worthy of publication, although it still requires some effort on the part of the authors. I therefore urge the authors to proceed with a revision of the manuscript based on the excellent comments and suggestions provided by Reviewer 2.**

We look forward to receiving your revised manuscript.

Kind regards,

Stefano Federici, Ph.D.

Academic Editor

PLOS ONE

Journal Requirements:

Additional Editor Comments :

After reading the revised version of the manuscript and based on the comments provided by Reviewer 2, I believe that the paper is worthy of publication, although it still requires some effort on the part of the authors. I therefore urge the authors to proceed with a revision of the manuscript based on the excellent comments and suggestions provided by Reviewer 2.

Reviewers' comments:

Reviewer's Responses to Questions

**Comments to the Author**

1. If the authors have adequately addressed your comments raised in a previous round of review and you feel that this manuscript is now acceptable for publication, you may indicate that here to bypass the “Comments to the Author” section, enter your conflict of interest statement in the “Confidential to Editor” section, and submit your "Accept" recommendation.

Reviewer #2: (No Response)

2. Is the manuscript technically sound, and do the data support the conclusions?

Reviewer #2: Yes

3. Has the statistical analysis been performed appropriately and rigorously? 

Reviewer #2: I Don't Know

4. Have the authors made all data underlying the findings in their manuscript fully available?

Reviewer #2: Yes

5. Is the manuscript presented in an intelligible fashion and written in standard English?

Reviewer #2: Yes

6. Review Comments to the Author

Reviewer #2: This article used data from more than 50,000 currently married and cohabiting women aged 15-49 in 11 countries in SSA from the Demographic and Health Surveys to investigate the relationship between female genital mutilation and safe sex negotiation. The findings are socially relevant given the continuing issue of female genital mutilation in the region. Overall, the paper is well-written and the conclusions are supported by the results presented. However, there are areas where the manuscript can be improved:

1. Some slight errors can be found in the text. For instance, see line 80 and line 170.

2. The improved discussion in the end is helpful. However, it would help the reader to highlight the theorized connection between female genital mutilation and safe sex negotiation in the introduction. Another option could be to have a small theoretical framework after the introduction, which highlights how the two are hypothesized to be related. Again, which mechanisms are at stake?

3. It is argued that the association between female genital mutilation and women’s ability to negotiate for safer sex in sub-Saharan Africa is investigated. However, the focus is just on 11 countries in sub-Saharan Africa. Including the Multiple Indicator Cluster Surveys would make the analysis more robust and accurate to make conclusions about the entire region. This would benefit the cross-validation of the findings. I understand that this could be beyond the scope of this analysis, but I could not refrain from wondering whether the results would hold when including more countries in sub-Saharan Africa. Can the authors explain why they did not use the MICS data and state how many countries were excluded as a result?

4. To the reader, various components of the analysis are unclear. For instance, how is the weighting of the data being taken into account in the analysis? So, what exact weighting procedure was used? If the DHS weights are used, one is given more weight to surveys with more data (nationally representative for each country, but not across countries). See the DHS sampling manual: https://dhsprogram.com/pubs/pdf/DHSM4/DHS6_Sampling_Manual_Sept2012_DHSM4.pdf. The suggested approach is to de-normalize and renormalize according to the relevant measure. Next to that, it is unclear whether multilevel analysis was performed. This is not mentioned in the paper, however, the results (intraclass correlation) suggest that this was performed. Finally, can information regarding multicollinearity be provided?

5. Regarding Table 4, I am wondering why models 1 and 2 are being shown. Before, it was argued that model 3 was the best estimation. So, why is this not used to perform the country analysis? Also, look at PLOS ONE guidelines for statistical reporting, https://journals.plos.org/plosone/s/submission-guidelines.#loc-statistical-reporting. This suggests including the complete estimations with particular format requirements in the appendix.

6. In the discussion, it is argued that both female genital mutilation and safe sex negotiation are related to social norms. Could the authors take this into account in the statistical analysis?

7. The results in ‘prevalence of female genital mutilation and safer sex negotiation among women in sub-Saharan Africa’ do not align with the results mentioned in the abstract.

7. PLOS authors have the option to publish the peer review history of their article (what does this mean?). If published, this will include your full peer review and any attached files.

Reviewer #2: No

---

## [Author Response · Author response to Decision Letter 1]

31 Oct 2023

Reviewer #2: This article used data from more than 50,000 currently married and cohabiting women aged 15-49 in 11 countries in SSA from the Demographic and Health Surveys to investigate the relationship between female genital mutilation and safe sex negotiation. The findings are socially relevant given the continuing issue of female genital mutilation in the region. Overall, the paper is well-written and the conclusions are supported by the results presented. However, there are areas where the manuscript can be improved:

Response: Thank you.

1. Some slight errors can be found in the text. For instance, see line 80 and line 170.

Response: We have corrected these errors.

2. The improved discussion in the end is helpful. However, it would help the reader to highlight the theorized connection between female genital mutilation and safe sex negotiation in the introduction. Another option could be to have a small theoretical framework after the introduction, which highlights how the two are hypothesized to be related. Again, which mechanisms are at stake?

Response: Thank you for this observation. A theoretical perspective has been provided.

3. It is argued that the association between female genital mutilation and women’s ability to negotiate for safer sex in sub-Saharan Africa is investigated. However, the focus is just on 11 countries in sub-Saharan Africa. Including the Multiple Indicator Cluster Surveys would make the analysis more robust and accurate to make conclusions about the entire region. This would benefit the cross-validation of the findings. I understand that this could be beyond the scope of this analysis, but I could not refrain from wondering whether the results would hold when including more countries in sub-Saharan Africa. Can the authors explain why they did not use the MICS data and state how many countries were excluded as a result?

Response: Thank you. We understand that data on FGM is available in the MICS. However, the focus of this study is to use the standard DHS to ascertain the association between FGM and safe sex negotiation. Also, there were variations in the surveys years for the MICS and DHS, which we think could have affected the study. Hence the decision to use only the DHS. As at the time of this study, all the countries that had data on both FGM and safe sex negotiation as well as the covariates.

4. To the reader, various components of the analysis are unclear. For instance, how is the weighting of the data being taken into account in the analysis? So, what exact weighting procedure was used? If the DHS weights are used, one is given more weight to surveys with more data (nationally representative for each country, but not across countries). See the DHS sampling manual: https://dhsprogram.com/pubs/pdf/DHSM4/DHS6_Sampling_Manual_Sept2012_DHSM4.pdf. The suggested approach is to de-normalize and renormalize according to the relevant measure. Next to that, it is unclear whether multilevel analysis was performed. This is not mentioned in the paper, however, the results (intraclass correlation) suggest that this was performed. Finally, can information regarding multicollinearity be provided?

Response: We took into account weighting at the country and pooled levels. Thus, we de-normalised and renormalised the dataset before the final analysis. Multilevel regression analysis was used to examine the association between FGM and safer sex negotiation. We also checked for evidence of collinearity and there was no evidence of high collinearity among the variables. The minimum, maximum, and mean variance inflation factor were 1.00, 3.74, and 1.84, respectively.

5. Regarding Table 4, I am wondering why models 1 and 2 are being shown. Before, it was argued that model 3 was the best estimation. So, why is this not used to perform the country analysis? Also, look at PLOS ONE guidelines for statistical reporting, https://journals.plos.org/plosone/s/submission-guidelines.#loc-statistical-reporting. This suggests including the complete estimations with particular format requirements in the appendix.

Response: Thank you. We have attached the complete table indicating Model O, I, II, and III as a supplementary file. 

6. In the discussion, it is argued that both female genital mutilation and safe sex negotiation are related to social norms. Could the authors take this into account in the statistical analysis?

Response: Thank you for your observation. This was beyond the scope of the study, hence, it has been captured as a limitation. 

7. The results in ‘prevalence of female genital mutilation and safer sex negotiation among women in sub-Saharan Africa’ do not align with the results mentioned in the abstract.

Response: We have corrected this error.

---

## [Decision Letter · Decision Letter 2]

16 Nov 2023

PONE-D-21-27783R2Female genital mutilation and safer sex negotiation among women in sexual unions in sub-Saharan Africa: Analysis of Demographic and Health Survey dataPLOS ONE

Dear Dr. Adu,

Thank you for submitting your manuscript to PLOS ONE. After careful consideration, we feel that it has merit but does not fully meet PLOS ONE’s publication criteria as it currently stands. Therefore, we invite you to submit a revised version of the manuscript that addresses the points raised during the review process.

**Just one more small effort to make the manuscript suitable for publication. I invite the authors to address the issues raised by the Reviewer.**

We look forward to receiving your revised manuscript.

Kind regards,

Stefano Federici, Ph.D.

Academic Editor

PLOS ONE

Journal Requirements:

Additional Editor Comments :

Just one more small effort to make the manuscript suitable for publication. I invite the authors to address the issues raised by the Reviewer.

Reviewers' comments:

Reviewer's Responses to Questions

**Comments to the Author**

1. If the authors have adequately addressed your comments raised in a previous round of review and you feel that this manuscript is now acceptable for publication, you may indicate that here to bypass the “Comments to the Author” section, enter your conflict of interest statement in the “Confidential to Editor” section, and submit your "Accept" recommendation.

Reviewer #2: (No Response)

2. Is the manuscript technically sound, and do the data support the conclusions?

Reviewer #2: Yes

3. Has the statistical analysis been performed appropriately and rigorously? 

Reviewer #2: Yes

4. Have the authors made all data underlying the findings in their manuscript fully available?

Reviewer #2: Yes

5. Is the manuscript presented in an intelligible fashion and written in standard English?

Reviewer #2: Yes

6. Review Comments to the Author

Reviewer #2: Thank you for the response to the comments that were made. The paper is well-written and conclusions are supported by the results presented. Many of the comments made were addressed appropriately! Some additional minor remarks:

- One more minor error in line 215

- Regarding the MICS surveys, I am not sure whether I understand the arguments made. So, why was the DHS chosen in particular? And what do you mean by variations in the survey years for the MICS? Anyhow, explaining why the MICS were not used and the possible downslides of this could be discussed in the limitation parts of the article.

- Thank you for responding to several of the questions regarding to the analysis and adjustments made to the paper. However, the information about the weighting is still not included in the article itself. I think it would be good for transparency and potential reproducibility of the methods and results to add these to the article as well.

- The statement in line 247 is only true for particular countries as well.

7. PLOS authors have the option to publish the peer review history of their article (what does this mean?). If published, this will include your full peer review and any attached files.

Reviewer #2: No

---

## [Author Response · Author response to Decision Letter 2]

14 Jan 2024

Reviewer #2: 

Thank you for the response to the comments that were made. The paper is well-written and conclusions are supported by the results presented. Many of the comments made were addressed appropriately! Some additional minor remarks:

- One more minor error in line 215

Response: We have corrected this error.

- Regarding the MICS surveys, I am not sure whether I understand the arguments made. So, why was the DHS chosen in particular? And what do you mean by variations in the survey years for the MICS? Anyhow, explaining why the MICS were not used and the possible downslides of this could be discussed in the limitation parts of the article.

Response: Thank you. We used the standard DHS because tot all the countries have the MICS dataset. Additionally, the data collection process for the MICS and the variable measurement except for the sociodemographic characteristics differs from the standard DHS. Hence, our decision to use the standard DHS dataset. However, we have acknowledged this as a limitation.

- Thank you for responding to several of the questions regarding to the analysis and adjustments made to the paper. However, the information about the weighting is still not included in the article itself. I think it would be good for transparency and potential reproducibility of the methods and results to add these to the article as well.

Response: Thank you. We have described how the weighting methodology used in the study.

- The statement in line 247 is only true for particular countries as well.

Response: We have corrected this by adding the countries.

---

## [Decision Letter · Decision Letter 3]

5 Feb 2024

Female genital mutilation and safer sex negotiation among women in sexual unions in sub-Saharan Africa: Analysis of Demographic and Health Survey data

PONE-D-21-27783R3

Dear Dr. Adu,

We’re pleased to inform you that your manuscript has been judged scientifically suitable for publication and will be formally accepted for publication once it meets all outstanding technical requirements.

Kind regards,

Stefano Federici, Ph.D.

Academic Editor

PLOS ONE

Additional Editor Comments (optional):

Reviewers' comments:

Reviewer's Responses to Questions

**Comments to the Author**

1. If the authors have adequately addressed your comments raised in a previous round of review and you feel that this manuscript is now acceptable for publication, you may indicate that here to bypass the “Comments to the Author” section, enter your conflict of interest statement in the “Confidential to Editor” section, and submit your "Accept" recommendation.

Reviewer #2: All comments have been addressed

2. Is the manuscript technically sound, and do the data support the conclusions?

Reviewer #2: Yes

3. Has the statistical analysis been performed appropriately and rigorously? 

Reviewer #2: Yes

4. Have the authors made all data underlying the findings in their manuscript fully available?

Reviewer #2: Yes

5. Is the manuscript presented in an intelligible fashion and written in standard English?

Reviewer #2: Yes

6. Review Comments to the Author

Reviewer #2: (No Response)

7. PLOS authors have the option to publish the peer review history of their article (what does this mean?). If published, this will include your full peer review and any attached files.

Reviewer #2: No

---

## [Editor Report · Acceptance letter]

21 Mar 2024

PONE-D-21-27783R3 

PLOS ONE

Dear Dr. Adu, 

I'm pleased to inform you that your manuscript has been deemed suitable for publication in PLOS ONE. Congratulations! Your manuscript is now being handed over to our production team.

Kind regards, 

on behalf of

Prof. Stefano Federici 

Academic Editor

PLOS ONE